# Finding Patient Zero: Learning Contagion Source with Graph Neural Networks

## Abstract

Locating the source of an epidemic, or patient zero (P0), can provide critical insights into the infection's transmission course and allow efficient resource allocation. Existing methods use graph-theoretic centrality measures and expensive message-passing algorithms, requiring knowledge of the underlying dynamics and its parameters. In this paper, we revisit this problem using graph neural networks (GNNs) to learn P0. We establish a theoretical limit for the identification of P0 in a class of epidemic models. We evaluate our method against different epidemic models on both synthetic and a real-world contact network considering a disease with history and characteristics of COVID-19. We observe that GNNs can identify P0 close to the theoretical bound on accuracy, without explicit input of dynamics or its parameters. In addition, GNN is over 100 times faster than classic methods for inference on arbitrary graph topologies. Our theoretical bound also shows that the epidemic is like a ticking clock, emphasizing the importance of early contact-tracing. We find a maximum time after which accurate recovery of the source becomes difficult, regardless of the algorithm used.

## 1 Introduction

The ability to quickly identify the origin of an outbreak, or "finding patient zero", is critically important in the effort to contain an emerging epidemic. The identification of early transmission chains and the reconstruction of the possible paths of diffusion of the virus can be the difference between stopping an outbreak in its infancy and letting an epidemic unfold and affect a large share of a population. Hence, solving this problem would be instrumental in informing and guiding contact tracing efforts carried out by public health authorities, allowing for optimal resource allocation that can maximize the probability of an early containment of the outbreak. Disease spreading is modeled as a *contagion process* on a network Stroock & Varadhan (2007); Pastor-Satorras et al. (2015) of human-to-human interactions where infected individuals are going to transmit the virus by infecting (with a certain probability) their direct contacts. In general, contagion processes can capture a wide range of phenomena, from rumor propagation on social media to virus spreading over cyber-physical networks Centola & Macy (2007); Baronchelli (2018); Wang et al. (2013); Mishra & Keshri (2013). Therefore, learning the source of a contagion process would also have broader impact on various domains, from detecting sources of fake news to defending malware attacks.

Learning the *index case*, or patient zero (P0), is a difficult problem. In this paper, we model disease spreading as a contagion process (chains of transmissions) over a graph. The evolution of an outbreak is noisy and highly dependent on the graph structure and disease dynamics. In addition, in real-world epidemics, there is often a delay from the start of the outbreak to when epidemic surveillance and contact tracing starts. Hence, we might only observe the state of the graph at some intermediate times without access to the complete chains of transmission. Furthermore, due to its stochastic nature, the same source node might lead to different epidemic spreading trajectories. Finally, learning P0 from noisy observations of graph snapshots is computationally intractable and the complexity grows exponentially with the size of the graph Shah & Zaman (2011).

Most work in learning the dynamics of a contagion process Rodriguez et al. (2011); Mei & Eisner (2017); Li et al. (2018a) have focused on inferring the *forward* dynamics of the diffusion. In epidemiology, for example, Pastor-Satorras & Vespignani (2001) have studied learning the temporal dynamics of diseases spreading on mobility networks. The problem of learning the *reverse* dynamics

and identifying diffusion sources has been largely overlooked due to the aforementioned challenges. Two of the most notable exceptions in the area are "rumor centrality" Shah & Zaman (2011) for contagion processes on trees and Dynamic Message-passing (DMP) on graphs Lokhov et al. (2014) but both require as input the parameters of the spreading dynamics simulations.

Our goal is to provide fresh perspectives on the problem of finding *patient zero* using graph neural networks (GNNs) Gilmer et al. (2017). First, we conduct a rigorous analysis of learning P0 based on the graph structure and the disease dynamics, allowing us to find conditions for identifying P0 accurately. We test our theoretical results on a set of epidemic simulations on synthetic graphs commonly used in the literature Erdös et al. (1959); Albert & Barabási (2002). We also evaluate our method on a realistic co-location network for the greater Boston area, finding performance similar to the synthetic data. While collecting labeled data to train GNN to find P0 may not be possible, training GNN using simulations on real contact-tracing data can provide a fast method for inferring P0 and help with planning and resource allocation. To the best of our knowledge, our work is the *first* to tackle the patient zero problem with deep learning and to test the approach on a realistic contact network. In summary, we make the following contributions:

- We find upper bounds on the accuracy of finding patient zero in graphs with cycles, independent of the inference algorithm used.
- We show that beyond a certain time scale the inference becomes difficult, highlighting the importance of swift and early contact-tracing.
- We demonstrate the superiority of GNNs over state-of-the-art message passing algorithms in terms of speed and accuracy. Most importantly, our method is model agnostic and does not require the epidemic parameters to be known.
- We validate our theoretical findings using extensive experiments for different epidemic dynamics and graph structures, including a real-world co-location graph of the COVID-19 outbreak.

## 2  RELATED WORK

**Learning contagion dynamics**   Learning forward dynamics of contagion processes on a graph is a well studied problem area. For instance, Rodriguez et al. (2011); Du et al. (2013) proposed scalable algorithms to estimate the parameters of the underlying diffusion network, a problem known as network inference. Deep learning has led to novel neural network models that can learn forward dynamics of various processes including neural Hawkes processes Mei & Eisner (2017) and Markov decision processes-based reinforcement learning Li et al. (2018a). Learning forward contagion dynamics have also been intensively studied in epidemiology Pastor-Satorras & Vespignani (2001); Vynnycky & White (2010), social science Matsubara et al. (2012), and cyber-security Prakash et al. (2012). In contrast, research in learning the reverse dynamics of contagion processes is rather scarce. Influence maximization Kempe et al. (2003), for instance, finds a small set of individuals that can effectively spread information in a graph, but only maximizes the number of affected nodes in the infinite time limit. Our problem is more difficult as we care not just about the number of infected nodes, but which nodes were infected.

**Finding patient zero**   In order to find patient zero, we aim to learn the reverse dynamics of contagion processes. Shah & Zaman (2011) were among the first to formalize the problem on trees in the context of modeling rumor spreading in a network. Prakash et al. (2012); Vosoughi et al. (2017) studied similar problems for detecting viruses in computer networks. More recent advances proposed a dynamic message passing algorithm Lokhov et al. (2014) and belief propagation Altarelli et al. (2014) to estimate the epidemic outbreak source. Fairly recently, Fanti & Viswanath (2017) reduced the deanonymization of Bitcoin to the source identification problem in an epidemic and analyzes the dynamics properties. On the theoretical side, Shah & Zaman (2011); Wang et al. (2014) analyzed the quality of the maximum likelihood estimator and rumor centrality, but only for the simple SI model on trees. Antulov-Fantulin et al. (2015) found detectability limits for patient zero in the SIR model using exact analytical methods and Monte Carlo estimators. Khim & Loh (2016); Bubeck et al. (2017) proved that it is possible to construct a confidence set for the predicted diffusion source nodes with a size independent of the number of infected nodes over a regular tree. Our work provides fresh perspectives on the patient zero problem on general graphs based on the recent development of graph neural networks

**Graph neural networks**  Graph neural networks have received considerable attention (see several references in Bronstein et al. (2017); Zhang et al. (2018); Wu et al. (2019); Goyal & Ferrara (2018)). While most research is focused on static graphs, a few have explored dynamic graphs Li et al. (2018b); You et al. (2018); Kipf et al. (2018); Pareja et al. (2019); Trivedi et al. (2019). For example, Kipf et al. (2018) propose a deep graph model to learn both the graph attribute and structure dynamics. They use a recurrent decoder to forecast the node attributes for multiple time steps ahead. Trivedi et al. (2019) take a continuous-time modeling approach where they take the node embedding as the input and model the occurrence of an edge as a point process. Xu et al. (2020) propose a temporal graph attention layer to learn the representations of temporal graphs. However, most research is designed for link prediction tasks and none of these existing studies have studied the problem of learning the source of the dynamics on a graph.

## 3  CONTAGION PROCESS AND PATIENT ZERO

Finding patient zero means tracing the contagion dynamics back to its initial state and identifying the first nodes that started spreading. Here, we describe the disease dynamics on a network using Susceptible-Infected-Recovered (SIR) and Susceptible-Exposed-Infected-Recovered (SEIR) Kermack & McKendrick (1927) compartmental models that assume that infected individuals develop immunity once they recover from the infections.

### 3.1  CONTAGION PROCESSES ON NETWORKS

In the SIR model, the population is split into three compartments: susceptible ($S$) who are susceptible to infection by the disease; infected ($I$) who have caught the disease and are infectious; removed ($R$) who are removed from consideration after experiencing the full infectious period.

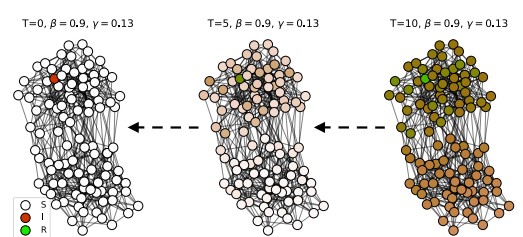

**Continuous time model**  For a contagion process on a graph $G$ with $N$ nodes, each vertex represents an individual who is in contact only with its neighbors. We can represent the graph using the adjacency matrix $A \in \mathbb{R}^{N \times N}$, where $A[i, j] = 1$ if two individuals are connected, 0

Figure 1: **Visualization of the patient zero problem:** uncover the original source (red node, left) given a future state of a contagion process (right).

otherwise. Let $S_i, I_i, R_i$ be the average probabilities of node $i$ being in each of the states, with $S_i + I_i + R_i = 1$. The SIR dynamics on a graph is given by Newman (2018):

$$\frac{dS_i}{dt} = -\beta \sum_j A_{ij} I_j S_i, \qquad \frac{dR_i}{dt} = \gamma I_i, \qquad \frac{dS_i}{dt} + \frac{dI_i}{dt} + \frac{dR_i}{dt} = 0. \qquad (1)$$

where $\beta$ is the infection rate per contact and $\gamma$ the recovery/death rate. The basic reproductive rate of a disease $R_0 \equiv \beta \lambda_1 / \gamma$ is defined as the number of secondary infections created by an index case in a fully susceptible population Keeling & Rohani (2011). Here $\lambda_1$ is the largest eigenvalue of $A$. The disease will spread and result in an epidemic if $R_0 > 1$.

**Discrete time model**  We can also use an equivalent discrete time SIR model. Let $x_i^t \in \{S, I, R\}$ be the state of node $i$ at time $t$. For a susceptible node $i$, its probability to become infected or removed at time $t + 1$ is

$$P(x_i^{t+1} = I | x_i^t = S) = 1 - \prod_j \left(1 - \beta A_{ij} I_i(t)\right), \qquad P(x_i^{t+1} = R | x_i^t = I) = \gamma. \qquad (2)$$

The SIR model doesn't account for the incubation period, where an individual is infected but not infectious. This is remedied by introducing an "exposed" (E) state, leading to the SEIR model. For a susceptible node $i$, the probability to enter the exposed state, and becoming infectious at time $t + 1$ is

$$P(x_i^{t+1} = E | x_i^t = S) = 1 - \prod_j \left(1 - \beta A_{ij} I_i(t)\right), \qquad P(x_i^{t+1} = I | x_i^t = E) = \alpha. \qquad (3)$$

An infected node eventually enters the removed state with probability $\gamma$, which is the same as SIR (2). (2) and (3) yield (1) for very small $\beta$ as $\prod_j (1 - \beta A_{ij} I_i) \approx \beta A_{ij} I_i$ (proof in supp. B).

**Finding patient zero**   Finding P0 can be formulated as a maximum likelihood estimation problem for SIR and SEIR models. Specifically, we observe a snapshot of the state of the nodes at time step $t$ as $\mathbf{x}^t := (x_1^t, \cdots, x_N^t)$, with each node's state $x_i^t \in \{S, E, I, R\}$. The problem of finding P0 is to search for a set of nodes $\mathcal{Z} = \{i | x_i^0 = I, i \in \{1, \cdots N\}\}$ which led to the observed state $\mathbf{x}^t$. A common approach is to find $\mathcal{Z}$ such that the likelihood of observing $\mathbf{x}^t$ is maximized:

$$\mathcal{Z}^\star = \operatorname{argmax}_{\mathcal{Z}, |\mathcal{Z}| \leq k} P(\mathbf{x}^t | \mathcal{Z}) \tag{4}$$

where $P(\mathbf{x}^t | \mathcal{Z})$ is the probability of observing $\mathbf{x}^t$ with $\mathcal{Z}$ being the P0 set. We assume the number of P0s is no larger than $k$. Estimating the original state of the dynamic system given the future states requires computing the conditional likelihood $P(\mathbf{x}^t | \mathcal{Z})$ exactly, which is intractable due to the combinatorics of possible transmission routes.

## 3.2   Fundamental limit of finding patient zero

The technical difficulty of finding P0 in SIR and SEIR stems from: (1) presence of cycles in graphs (higher-order transmission) (2) the removed state introducing additional uncertainty about temporal order of infections (3) uncertainty of the exact time step of the observed states. For SI dynamics (i.e. infection is permanent) on trees, existing theoretical results Shah & Zaman (2011); Khim & Loh (2016) have established upper bounds on the detection probability based on an estimator called "rumor centrality". For graphs with cycles, finding P0 becomes more elusive. We derive the fundamental limit considering the case where at time $t = 0$ one node, P0, is infected and all of the other nodes are susceptible.

**Ambiguity of patient zero on cyclic graphs**   For graph with cycles, if a cycle is embedded within the infected subgraph, it will reduce the accuracy of predicting P0 because multiple spreading scenarios can lead to the same infection pattern in the cycle. For instance, take a 3-regular tree where the infection has started from the root and spread to some level. If any two branches are connected to make a cycle, both branches become equally likely to have spread the disease downstream. Therefore, having cycles in the infected subgraph reduces the accuracy of finding P0. If $O(N)$ nodes in a graph with cycles are infected, cycles will likely interfere with finding P0. Based on this observation, the following theorem estimates the time horizon beyond which finding P0 becomes difficult.

**Theorem 1** (Time Horizon). *Assume SIR dynamics* (1) *on a connected graph of $N$ nodes, starting with a single patient zero. Denoting the adjacency matrix by $A$ and its largest eigenvalue by $\lambda_1$, the average infection probability, both over nodes and choice of patient zero, $\langle I(t) \rangle \equiv \langle \sum_i I_i(t)/N \rangle_{\mathrm{P0}}$ becomes $O(1)$ after $t_{\max}$ time steps given by*

$$t_{\max} \sim \frac{\log N}{\gamma(R_0 - 1)}, \qquad R_0 \equiv \frac{\beta \lambda_1}{\gamma} \tag{5}$$

*Proof:* The proof (in Supp B.5.1) follows from the exponential behavior of (1) at small $t$ (Newman, 2018)

$$I_i(t) \approx \sum_j \exp\left[t(\beta A - \gamma \mathbf{I})\right]_{ij} I_j(0) \approx \exp\left[(\beta \lambda_1 - \gamma)t\right] \left(\psi^{(1)} \cdot I(0)\right) \psi_i^{(1)}, \tag{6}$$

and the leading eigenvector being positive $\sum_i \psi_i^{(1)} \geq \left\|\psi^{(1)}\right\|_2 = 1$ (Perron-Frobenius theorem).   $\square$

The maximum detection accuracy of P0 on a connected graph would decrease with increasing number of cycles in the infected subgraph, $G_I$. In special cases we can estimate the penalty to the accuracy due to cycles. For example, in connected Erdős–Rényi (ER) random graphs Erdös et al. (1959), where each edge has independent an probability $p$, the number of cycles can be estimated. To get a conservative estimate, we focus on triangles, as they are the most prevalent cycles in ER, to get an upper bound for the accuracy in the following theorem. This ignores cases where the presence of triangles causes downstream error or the error arising from other types of cycles.

**Theorem 2** (Detection Accuracy). *In contagion process on a connected random graph $G$, with edge probability $p$ and with infected subgraph $G_I$, the prediction accuracy for P0 is bounded from above*

$$P_{\max} < \frac{1}{3} + \frac{2}{3}(1-p)^{\binom{|G_I|p}{2}} \tag{7}$$

The proof (supp. B.5.2) follows from estimating number of triangles in subgraph $G_I$ of a dense ER graph and noting each triangle can drop the accuracy of P0 to $1/3$.

Figure 2 shows an example of how this upper bound behaves for different values of $R_0$. The graph is a uniformly connected ER of $N = 100$, $p = 2 \log N / N$ and with $\gamma = 0.4$. In conclusion, on graphs with cycles, we expect finding P0 after a time $t_{\max} \sim O(\log N)$ to become difficult. This suggests that to find P0 contact-tracing must be done promptly and in early stages.

## 4    FINDING PATIENT ZERO WITH GRAPH NEURAL NETWORKS

We propose using GNNs for finding P0 and show that we can improve significantly upon state-of-the-art methods, e.g. DMP. Moreover, using GNNs gives us the distinct advantage that they are model-agnostic and do not require access to the epidemic dynamics parameters or the time $t$ of the graph snapshot. Our goal is not to propose a novel graph neural network architecture, but to understand the trade-off between different probabilistic inference methods in the context of contagion dynamics. Before we discuss our GNN solution, we briefly review Dynamic Message Passing.

**Dynamic Message Passing**    DMP Lokhov et al. (2014) estimates the probability of every node being the P0 in the SIR model using message-passing equations and approximates the joint likelihood with a mean-field time approach by assuming the following factorization:

$$P(\mathbf{x}^t|\mathcal{Z}) \approx \prod_{i, x_i^t = S} P(x_i^t|\mathcal{Z}) \prod_{j, x_j^t = I} P(x_j^t|\mathcal{Z}) \prod_{k, x_k^t = R} P(x_k^t|\mathcal{Z}) \tag{8}$$

The algorithmic complexity of the DMP equations over a graph with $N$ nodes and $t \leq T$ diffusion steps is $O(TN^2\langle k \rangle)$ where $\langle k \rangle$ is the average degree of the graph. Furthermore, DMP requires providing the SIR epidemic parameters and the time $t$ of the graph snapshot before performing inference. For comparison, on a connected random graph, $\langle k \rangle > \log N$, yielding $> O(TN^2 \log N)$ time complexity for DMP. A trained GNN with $l$ layers (2–8 in our case) has complexity $O(N^2l)$ in the inference step and does not require inputting the model parameters. This makes it harder to scale DMP for large or dense graphs. DMP is proven to be exact on trees, e.g. Kanoria et al. (2011), and has been used on more general graphs with reasonable success. In practice, we find that GNN inference can be 50–100 times faster than DMP, as DMP has to exhaustively infer the marginals of $S, I, R$ states for each node for each time $t \geq 0$. DMP is trying to accurately infer the distribution over the nodes and auto-regressively infer P0, whereas GNN does not infer node states at each time $t$ and tries to directly infer P0. On the other hand, DMP is an unsupervised inference algorithm and does not "learn", while GNN is provided with labels in a standard supervised setting.

**Relation between Contagion Dynamics and GNNs**    Our use of GNNs for finding P0 is motivated by the fact that the contagion dynamics (1) are a special case of Reaction-Diffusion (RD) processes on graphs Colizza et al. (2007) which is structurally equivalent to GNNs, as shown next.

**Proposition 1.** *Reaction-diffusion dynamics on graphs is structurally equivalent to the message-passing neural network ansatz.*

Denoting $p_i^\mu(t) \equiv P(x_i^t = \mu)$ of node $i$ being in states such as $\mu \in \{S, I, R\}$ or $\mu \in \{S, E, I, R\}$ at time $t$, a Markovian reaction-diffusion dynamics can be written as

$$p_i^\mu(t+1) = \sigma\left(\sum_j F\left(\mathcal{A}_{ij} \cdot h(p_j)^\mu\right)\right), \qquad h_a(p_i)^\mu = \sigma\left(\sum_\nu W_{a,\nu}^\mu p_i^\nu + b^\mu\right) \tag{9}$$

where $\mathcal{A}_{ij}^a = \theta(A_{ij})f(A)_{ij}$ with $\theta(\cdot)$ being the step function and $\sigma(\cdot)$ a nonlinear function. RD on graphs is structurally equivalent to Message-passing Neural Networks (MPNN) Gilmer et al. (2017), as RD involves a message-passing step and a node-wise interaction among features (Supp. B.3),

same as MPNN. We choose the simpler architecture of GCN as in (10) instead of general MPNN. Finding P0 requires learning the backward dynamics of RD, which seems to require the inverse of the propagation rule (PR). Yet, each node can only get infected by its neighbors, so even the backward dynamics requires message passing over the same adjacency matrix and should again have the structure of RD.

## 4.1 LEARNING WITH GRAPH NEURAL NETWORKS

We employ a state-of-the-art GNN design, suggested by Dwivedi et al. (2020). We make several modifications to the model architecture to fit our problem. Given one-hot encoded node states $x_i^t \in \{0,1\}^M$ as the GNN input, where $M$ is the number of states and where the states are either $\{S, E, I, R\}$ or $\{S, I, R\}$, we first apply a linear transformation $h_i^{(0)} = U x_i^t$ with $U \in \mathbb{R}^{C \times M}$. Denote the output of layer $l$ by $h_i^{(l)}$, where $i$ is the node index. We use graph convolutional network (GCN) Kipf & Welling (2016) in each layer $g(h) = \sigma_g (f(A) \cdot h \cdot W + b)$, where $W \in \mathbb{R}^{C \times C}$, $b \in \mathbb{R}^C$ and $f$ is called the propagation rule in GCN. We use $f(A) = D^{-1/2} A D^{-1/2}$ for the propagation rule, where $D_{ij} = \delta_{ij} \sum_k A_{ik}$ is the degree matrix. To include features of the central node, instead of adding self-loops, we use residual connections between GCN layers and notice a significant increase in model performance. The action of these higher GNN layers is given by

$$h_i^{(l+1)} = h_i^{(l)} + \sigma\left(\texttt{BN}(g(h_i^{(l)}))\right), \qquad\qquad y_i = P \cdot \texttt{ReLU}(Q \cdot h_i^{(L)}) \qquad (10)$$

where $L$ is the number of layers and the output layer is parameterized by $Q \in \mathbb{R}^{D \times D}$ and $P \in \mathbb{R}^{1 \times D}$ to generate $y_i \in \mathbb{R}$, representing the probability that node $i$ is P0. $\texttt{BN}(\cdot)$ denotes Batch Normalization and $\sigma(\cdot)$ is a leaky-relu nonlinear activation function.

## 5 EXPERIMENTS

We perform extensive studies on performance of our GCNs in finding P0 in SIR and SEIR dynamics over synthetic graphs with various graph topologies. In addition, we generate synthetic epidemic outbreaks on top of a real world co-location network using a SEIR compartmental model that is calibrated to simulate a contagion process with characteristics similar to a COVID-19 outbreak.

**Experimental Setup** We compare the performance of DMP Lokhov et al. (2014) and different variants of GCNs, following the architecture we described in sec. 4.1:

• `DMP`: Dynamic Message Passing algorithm
Lokhov et al. (2014), we sample a graph snapshot $O$ at time $t$ with each node having a state $x_i^t \in \{S, I, R\}$, and select the node $i$ that has the highest likelihood of generating $O$, that is P0= $\text{argmax}_i P(O|x_i^0 = I)$.
• `GCN-S`: symmetric GCN Kipf & Welling (2016) $f(A) = D^{1/2} A D^{1/2}$, `GCN-R`: random walk $f(A) = D^{-1} A$, `GCN-M`: mixture of propagation rules $f(A) = A || D^{1/2} A D^{1/2}$
• `GAT`: Graph Attention Network Veličković et al. (2017)

We train our models using DGL Wang et al. (2019) with a PyTorch backend. The task is to predict the probability for each node being P0 given a single graph snapshot. We report performance averaged over four random seeds. Details on training and hardware are given in Supp. A.2 We additionally report inference run times.

**Evaluation Metrics** We use top-1 accuracy to understand the effectiveness of our method. However, due to the ambiguity of detecting patient zero, as elaborated in our theoretical analysis, top-1 accuracy may not be the only evaluation measure to be relied upon. Therefore we also calculate the normalized rank defined by $R_t = 1 - \frac{1}{|D_t| N} \sum_{u \in D_t} r_u$ where $D_t$ is the set of test samples at time $t$, N is the size of the graph and $r_u$ is the index of the ground truth P0 in the reverse-sorted probability distribution. Normalized rank is a retrieval metric that tells us how high the correct patient zero was in the learned output distribution. It demonstrates the quality of the output distribution in learning the stochastic dynamics and helps us understand how high was P0 ranked even if it was not ranked the first.

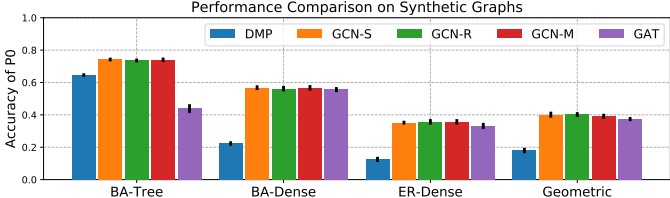

| **Inference times** | | | |
|---|---|---|---|
| **Dataset** | DMP | GCN | GAT |
| BA-Tree | 14.40 hr | 3.89s | **3.18s** |
| BA-Dense | 77.04 hr | **4.91s** | 8.19s |
| ER-Dense | 71.77 hr | **4.93s** | 9.66s |
| Geometric | 70.35 hr | **5.34s** | 10.87s |

Figure 3: Mean Prediction accuracy/speed comparison for different methods for the test set over $T = 30$ steps and $R_0 = 2.5$. The time to perform inference over the test set for different models have been listed above. Note that the time taken by GCN represents the mean time taken by GCN variants. We observe that GNNs beat DMP by a large margin both in terms of speed and accuracy. The similar performance of the three GNN models may be due to approaching the theoretical limits.

## 5.1 EXPERIMENTS WITH SYNTHETIC NETWORKS

We first experiment on three synthetic graph models: ER random graph, Barabási-Albert (BA) graph Albert & Barabási (2002) and Random Geometric Graph (Geometric or RGG) Dall & Christensen (2002), and use NDLib (Rossetti et al., 2017) to simulate SIR and SEIR epidemic dynamics on the graph (Supp. A.1). We pick a random P0 seed node at time $t = 0$ and run S(E)IR a fixed number of steps $T$. The epidemic parameters $(\alpha, \beta, \gamma)$ are chosen such that we can vary $R_0$ to study model performance. We set $\gamma = 0.4$ and $\beta = R_0\gamma/\lambda_1$ where $\lambda_1$ is the largest eigenvalue of the graph. For SEIR, we set $\alpha = 0.5$. We generate $20,000$ simulations and use $80 - 10 - 10$ train-validation-test split. For each sample we select $t \in \{1, \cdots T\}$ uniformly at random and try to predict P0 at time $t = 0$ given the graph adjacency matrix $A$ and node features $x_i^t$.

We first compare the top-1 prediction accuracy in SIR and SEIR for different models averaged over $1 \le t \le T$. Fig 3 compares the prediction accuracy and average inference time for different models. We can see that GNN-based models outperform the baseline DMP both in accuracy and efficiency. We also want to note that the training time for GNNs is under 7 hours, significantly less than the fastest DMP run of 14.40 hr. It is also worth emphasizing that DMP requires explicit input of $\beta, \gamma$ and $t$ while GNNs are model agnostic.

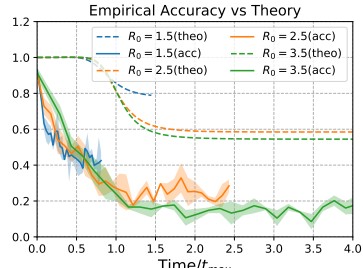

Figure 2: Theoretical upper bound on accuracy (dashed line) vs experimental (solid line) on ER for varying $R_0$. While accuracy drops below the theoretical limit at $t \sim t_{max}$.

To validate our theory, we plot the theoretical accuracy upper-bound and the empirical accuracy obtained from GNN in Fig. 2. We note that the time scale and drop in accuracy are consistent with our theoretical results on $t_{max}$ (28) and the upper bound on accuracy $P_{max}$ (7). Combined with the fact that all our GNN models have comparable accuracies, this suggests that our GCN-based models may be approaching the fundamental limits we described in 3.2.

Fig. 4 shows the trend of accuracy decay over the time steps $t$ for different graph structures and $R_0$ values. As expected, the accuracy is highest on a tree and when $t$ is small. In graphs with cycles (BA-Dense, ER-Dense, and Geometric) we also observe a nontrivial drop in accuracy which depends both on $t$ and $R_0$. For SIR we observe a drop in accuracy as a function of $R_0$ and time, consistent with our theoretical upper bound. The decay is slower for SEIR as the latent stage adds a delay to the spread of the epidemic. The normalized rank of P0 remains high even over longer time horizons, indicating that P0 could be narrowed down to small subset of the population with impressive accuracy.

## 5.2 BOSTON CO-LOCATION NETWORK AND COVID-19 EPIDEMIC TRAJECTORY

Our real-world dataset consists of a co-location graph and simulations of an epidemic with the natural progression of COVID-19. The co-location graph is constructed using the Cuebiq data[1] for two weeks from 23 March, 2020 to 5 April, 2020 ($N = 384,590$ nodes). To reduce computational costs, we sample a subgraph with $N = 2,689$ nodes and $|E| = 30,376$ edges while maintaining the

---

[1]https://www.cuebiq.com/about/data-for-good/ derived from Klein et al. (2020a;b)

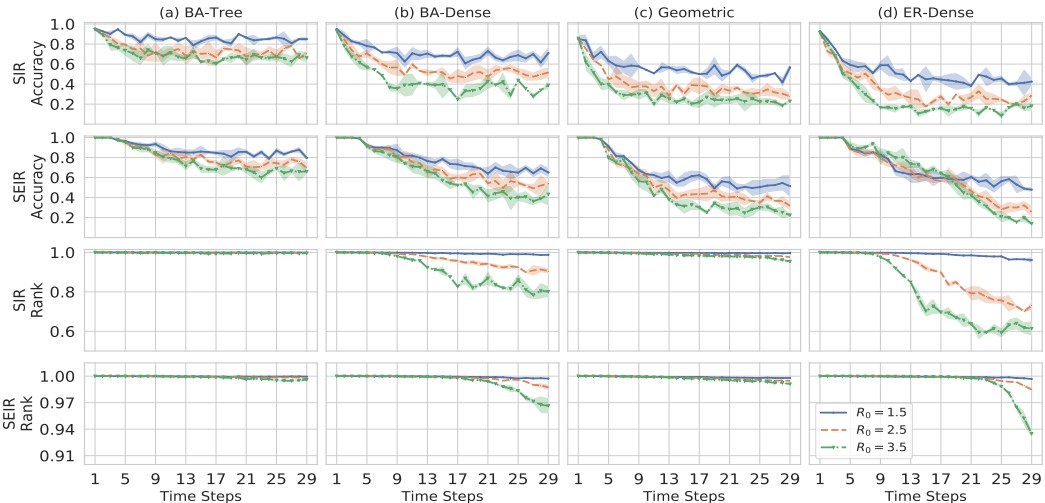

Figure 4: Performance of GCN-S for SIR and SEIR epidemic dynamics as top-1 accuracy over the test set. The top-1 recovery accuracy of P0 vs time (first and second row) and normalized rank of P0 (third and fourth row) for different graph topologies with varying $R_0$ values. Note that in BA-Tree (a), which is a tree, the accuracy remains fairly high in both SIR and SEIR, consistent with existing literature, and confirming that cycles significantly reduces accuracy of P0.

degree distribution and connectivity patterns of the original graph. For the epidemic simulations, we run a modified SEIR model with asymptomatic infectious states on the co-location graph with $R_0$ resembling COVID-19 Chinazzi et al. (2020) and accordingly set $R_0 = 2.5$. Each simulation contains 1 patient zero, selected uniformly at random. The simulation is run for 50 days. We create a dataset with $10,000$ samples and an $80 - 10 - 10$ train-validation-test split (supp. C).

The top-k accuracy performance over different days when the graph snapshot was observed are shown in Fig. 5a. We can see that the top-1 accuracy falls steadily over time, the top-(10, 20) accuracy remains fairly high for the first two weeks suggesting that we can retrieve P0 in the most likely 20 nodes out of a total $2,689$ candidates.

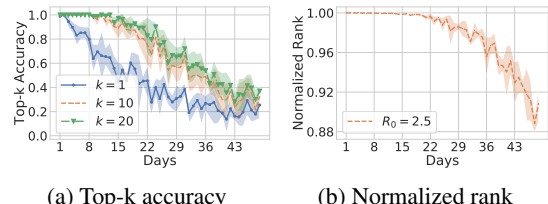

(a) Top-k accuracy          (b) Normalized rank

Interestingly, while the top-1 accuracy decreases significantly, degrading by 50% after 14 days, using normalized rank, the model can narrow down the set of patient zeros accurately even later in the epidemic, as shown in Fig 5b. For the normalized rank, P0 can be recovered fairly accurately in the first two weeks of the

Figure 5: Performance on Boston co-location network with simulations following the natural history of COVID-19. Shown here are the top-k accuracy and normalized rank.

epidemic. These results highlight an important trade-off between accurately determining patient zero and retrieving the general infected region.

## 6  CONCLUSION

We study contagion dynamics on a graph using graph neural networks (GNNs) to learn the reverse dynamics of contagion processes and predict patient zero. We evaluate our method against different epidemic models on both synthetic and a real-world contact network with a disease with the natural history and characteristics of COVID-19. We observe that GNNs can efficiently infer the source of an outbreak without explicit input of dynamics parameters. Most notably, GNN accuracy approaches our predicted theoretical upper bound, indicating that further architecture refinements may not improve performance significantly. In addition, GNN is over 100x faster for inference than classic methods for arbitrary graph topologies. Extensions of this work may include learning using sequences of graph snapshots, as well as allowing a set of patient zeros.

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

# A  APPENDIX

## A.1  DATASET DETAILS

We use three graph models: ER random graph, Barabási-Albert (BA) graph Albert & Barabási (2002) and Random Geometric Graph (Geometric or RGG) Dall & Christensen (2002). The density of all three ($|E|/\binom{N}{2}$) is adjustable, but BA can produce exact trees. Fixing the number of nodes to $N = 1,000$, we first obtain one random instance of tree BA, and dense BA, ER and Geometric graphs with $|E| \approx 10,000$ using the NetworkX library (NetworkX developer team, 2014) and then use NDLib (Rossetti et al., 2017) to simulate SIR and SEIR epidemic dynamics on the graph (supp. A.1). For each sample graph, we pick a P0 seed node $i$ at random to be the patient zero at time $t = 0$ and then we run S(E)IR a fixed number of steps $T$. The epidemic parameters $(\alpha, \beta, \gamma)$ are chosen such that we can vary $R_0$ to study model performance. We set $\gamma = 0.4$ and $\beta = R_0 \gamma / \lambda_1$ where $\lambda_1$ is the largest eigenvalue of the graph. For SEIR, we set $\alpha = 0.5$. We generate $20,000$ simulations and use $80 - 10 - 10$ train-validation-test split. For each sample we select $t \in \{1, \cdots T\}$ uniformly at random and try to predict P0 at time $t = 0$ given the graph adjacency matrix $A$ and node features $x_i^t$.

Table 1 describes the details of the synthetic datasets.

Table 1: Description of the sampled graph statistics

| Dataset | # of Nodes | # of Edges | Density | Diameter |
|---|---|---|---|---|
| BA-Tree | 1,000 | 999 | 0.99 | 19 |
| BA-Dense | 1,000 | 9,900 | 9.90 | 4 |
| Geometric | 1,000 | 9,282 | 9.28 | 21 |
| ER-Dense | 1,000 | 9,930 | 9.93 | 4 |

## A.2  TRAINING AND HARDWARE

We train the model with an ADAM optimizer for 150 epochs with an initial learning rate of $0.003$ and decay the learning rate by $0.5$ when the validation loss plateaus with a patience of 10 epochs. We perform hyperparameter tuning over a validation set with a random search strategy. We sweep over the hyperparameter space and track our experiments using Weights and Biases Biewald (2020) choosing the model with the lowest validation error. We run our experiments on Nvidia 2080Ti GPUs and report performance averaged over 4 random seeds.

## A.3  HYPER-PARAMETER DETAILS

Table 2: Description of hyper-parameters used. All of our models have been trained with 4 random seeds. The initial learning rate is mentioned in the table below and additionally we decay the learning rate by 0.5 with a patience of 10 epochs when the validation error plateaus. Note that GAT had 4 attention heads and has been trained with 5 layers due to a limitation on GPU memory.

| Hyperparameters | GCN-S | GCN-R | GCN-M | GAT |
|---|---|---|---|---|
| Number of Epochs | 150 | 150 | 150 | 150 |
| Batch Size | 128 | 128 | 128 | 32 |
| GNN Hidden Dim | 128 | 128 | 128 | 128 |
| Dropout | 0.265 | 0.265 | 0.265 | 0.265 |
| Number of GNN Layers | 10 | 10 | 10 | 5 |
| Initial Learning Rate | 0.0033 | 0.0033 | 0.0033 | 0.004 |

## A.4  NOTES ON DMP IMPLEMENTATION

We include DMP Lokhov et al. (2014) as a baseline against our proposed GNN based method. As DMP does not have code that is publicly available, we implemented DMP using Python for a fair

comparison with GNNs. Accordingly, our implementation of DMP uses DGL Wang et al. (2019) which enables us to vectorize belief propagation (BP) and marginalization and now it runs in parallel for all nodes.

Given a graph $G(V, E)$, we observe $O^t$ as the state of the graph with nodes $i \in V$. DMP employs MLE estimation to determine the node $i_{P0}$ that may have led to the observed snapshot $O$. For a single sample in our dataset $D$, we use algorithm 1. In order to implement DMP efficiently, we implemented it as a message-passing on a graph using DGL. We sequentially initialize node and edge features for all node $i$ and then as we obtain $N = |V|$ set of graphs with node $i$ acting as P0 in $G_i$. DMP then allows us to obtain $i = \arg\max_i P(O|i)$. The advantage of our implementation then is that we can process all $N$ graphs in parallel as if it were one large graph with $N^2$ nodes and $E^2$ edges thanks to DGL's support for batching graphs. A salient feature of using DGL is that the message passing framework allows us to additionally process all the nodes and edges for a single time step $t$ in parallel. The nature of BP algorithms do not allow us to do away with the for-loop over time $\mathbf{t}$ and that remains the only sequential aspect of our implementation. Finally, we use algorithm 1 to process each sample in our test set sequentially. It should be noted that we can further vectorize over a batch of samples in our test set. However, the memory required for DMP is $O(bN^2E^2)$ with $b$ being the size of the batch and so memory requirements quickly blow up. Accordingly, we leave this aspect of implementation for future work.

---

**Algorithm 1:** Dynamic Message Passing given graph $G$, snapshot $O$ and time $\mathbf{t}$

---
**for** $i \in V$ **do**
    set node $i$ to be P0
    initialize node features and edge features with eq (12, 13) in DMP;
    **for** *(t = 0; t < t; t = t + 1)* **do**
        **for** $e \in E$ **do**
            perform message passing with eq (15, 16, 17) in DMP
    **for** $j \in V$ **do**
        marginalize and update node states with eq (18, 19, 20) in DMP.
    Calculate $P(O|i)$ with eq 21 in DMP.
**return** $i = argmax_i P(O|i)$

---

## A.5    EFFECT OF VARYING NUMBER OF GCN-S LAYERS ON TOP-1 ACCURACY

Fig. 6 shows the top-1 accuracy of P0 of the GCN-S model for varying number of layers. We do not observe a significant effect coming from the number of layers. This may be due to the accuracy limitations with $t_{\max}$ and cycles affecting all the models equally, and superseding other effects such as the diameter of the graph. Another possible reason may be that the 20,000 samples on a graph of 1,000 nodes has many repetitions of the same P0, resulting in both shallow and deep models memorizing patterns.

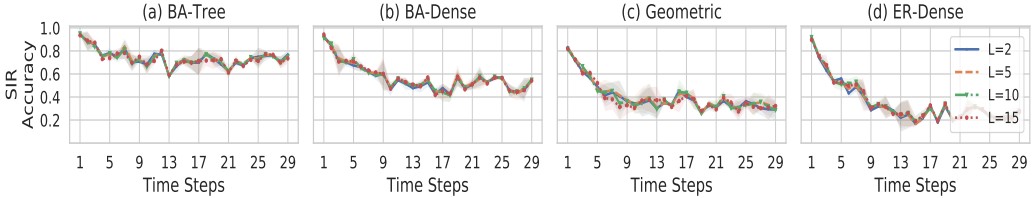

Figure 6: Performance of GCN-S for SIR epidemic dynamics as top-1 accuracy over the test set with varying number of layers.

## B  THEORETICAL ANALYSIS

### B.1  EARLY STAGE EVOLUTION OF SIR AND SEIR

The SIR equation on a graph are

$$\frac{dS_i}{dt} = -\beta \sum_j A_{ij} I_j S_i, \qquad \frac{dR_i}{dt} = \gamma I_i, \qquad \frac{dS_i}{dt} + \frac{dI_i}{dt} + \frac{dR_i}{dt} = 0. \qquad (11)$$

In very early stages, when $t \ll 1/\gamma$ and $\sum_i I_i + R_i \ll N$, we have $S_i \approx 1$ and we have exponential for $I_i$ because

$$\frac{dS_i}{dt} = -\frac{dI_i}{dt} - \frac{dR_i}{dt} = -\frac{dI_i}{dt} - \gamma I_i$$

$$\frac{dI_i}{dt} = \beta \sum_j A_{ij} I_j S_i - \gamma I_i \approx \sum_j (\beta A_{ij} - \gamma \delta_{ij}) I_j$$

$$I_i(t) \approx \sum_j (\exp[t (\beta A - \gamma \mathbf{I})])_{ij} I_j(0) \qquad (12)$$

Expanding this using the eigen-decomposition $A = \sum_i \lambda_i \psi_i \psi_i^T$ yields eq. (29).

### B.2  TRANSITION PROBABILITIES

More generally, when the graph is weighted, the probability of susceptible node $i$ getting infected depends on $A_{ij}$ and the probability of node $j$ being in the infected state. For brevity, define $p_i^\mu(t) \equiv P(x_i^t = \mu)$, with $\mu \in \{S, I, ..., R\}$. The infection probability in SIR (2) can be written as

$$P(x_i^{t+1} = I | x_i^t = S) = 1 - \prod_j (1 - \beta A_{ij} p_j^I) = \beta \sum_j A_{ij} p_j^I - \beta^2 [A p^I]^2 + O(\beta^3). \qquad (13)$$

### B.3  REACTION DIFFUSION FORMULATION

For brevity, define $p_i^\mu(t) \equiv P(x_i^t = \mu)$. In a network diffusion process the assumption is that node $i$ can only be directly affected by state of node $j$ if there is a connection between them, i.e. if $A_{ij} \neq 0$. This restriction means that the general reaction-diffusion process on a graph has the form

$$F_a(A; p)_i^\mu \equiv \sum_j f_a \Big( g_a(A)_{ij} h_a(p_j)^\mu \Big) \qquad (14)$$

$$p_i^\mu(t+1) = F(A; p(t))_i^\mu = \sigma \left( \{F_a(A; p(t))_i^\mu\} \right) \qquad (15)$$

With

$$g_a(A)_{ij} = \theta(A_{ij}) \tilde{g}_a(A)_{ij} \qquad h_a(p_i)^\mu = \sigma_a \left( \sum_\nu W_{a,\nu}^\mu p_i^\nu + b_a^\mu \right) \qquad (16)$$

where $\theta(\cdot)$ is the step function and $\sigma_a(\cdot)$ a nonlinear function. In regular diffusion on a graph, we have two states $S, I$ and diffusion is changing the $S \to I$ state. The probability $P_{ij} \equiv P(x_i^{t+1} = I | x_j^t = S)$ of node $i$ getting infected at $t + 1$, given node $j$ was in the infected state at time $t$, can be expressed in the form of is determined by the adjacency matrix $A_{ij}$ because node $j$ can only infect its neighbors. The infection probability is given by $p_i^I(t+1) = \beta A_{ij} p_j^I(t)$ and $p_i^S = 1 - p_i^I$. Hence, for diffusion

$$f_1(x) = x \qquad g_1(A) = \beta A, \qquad h_1(p_j)^\mu = \sum_\nu \delta_I^\mu \delta_\nu^I p_j^\nu. \qquad (17)$$

In regular diffusion there is no condition on the target node $i$ and even if it is in the $I$ state the dynamics is the same. In the SI model, however, the infection only spreads to $i$ if it is in the $S$ state. Thus, we have to multiply the dynamics by $p_i^S \equiv P(x_i^t = S)$ which yields

$$p_i^I(t+1) = \beta A_{ij} p_j^I(t) p_i^S. \qquad (18)$$

This can still be written as (9) by adding the extra functions

$$f_2(x) = x, \qquad g_2(A) = I, \qquad h_2(p_j)^\mu = \sum_\nu \delta_S^\mu \delta_\nu^S p_j^\nu \qquad (19)$$

and having

$$p_i(t+1)^I = F_1(A; p(t))_i^I F_2(A; p(t))_i^S \qquad (20)$$

where $F_a = f_a(g_a \cdot h_a)$ are as in (14). More complex epidemic spreading models such as SIR and SEIR can also be written in a similar fashion. In SIR and SEIR the rest of the dynamic equations are linear and do not involve the the graph adjacency $A$ at all, meaning $g_a(A) = I$ in the rest of the equations.

## B.4 DISCRETE TIME AGENT-BASED SIR AS A REACTION DIFFUSION SYSTEM

The agent-based models (2) and (3), which correct for double-counting of infection from multiple neighbours, are sometimes written as

$$P(x_i^{t+1} = I | x_i^t = S) = 1 - (1 - \beta)^{\xi_i}, \qquad (21)$$

where $\xi_i$ is the total number of neighbors $j$ of $i$ which are infected, meaning $x_j^t = I$. We will first show that this is a special case of the form given in our paper. First, note that in (2) the terms can also be written as

$$(1 - \beta)^{\xi_i} = \prod_j \left( 1 - \beta \delta_{x_j^t, I} \right) \qquad (22)$$

In the probabilistic model, we have to replace the strict condition of $j$ being in the $I$ state with its probability, so $\delta_{x_j^t, I} \to P(x_j^t = I) = p_j^I(t)$.

$$P(x_i^{t+1} = I | x_i^t = S) = 1 - \prod_{j \in \partial_i} \left( 1 - \beta \hat{A}_{ij} p_j^I \right) \qquad (23)$$

and for small $\beta$ yield

$$P(x_i^{t+1} = I | x_i^t = S) = \beta \sum_j \hat{A}_{ij} p_j^I - \beta^2 \sum_{j,k} \hat{A}_{ij} p_j^I \hat{A}_{ik} p_k^I + O(\beta^3) \qquad (24)$$

which yields the simplified equation $p_i(t+1)^I = p_i^S(t) \sum_j \beta \hat{A}_{ij} p_j^I(t)$. Note that if the infection rate per time step $\beta$ is large $\beta \sum_j \hat{A}_{ij} p_j$ can exceed 1, rendering (24) inconsistent with $p_i^I$ being probabilities. Both (23) and (24) both can be written in the form of RD (15) and (9). We utilize the $h_1, g_1$ and $h_2, g_2$ found for diffusion (17) and SI (19)

$$F_1(A; p)_i^\mu = \sum_j \log \left( 1 - \beta \hat{A}_{ij} h_1(p_j)^\mu \right) \qquad F_1(A; p)_i^\mu = h_2(p_i)^\mu \qquad (25)$$

and defining the probability as

$$p_i^I(t+1) = F_1_i^S \left( 1 - \exp\left[ F_2_i^I \right] \right) = p_i^S(t) \left( 1 - \prod_j \left( 1 - \beta A_{ij} p_j^I(t) \right) \right)$$

$$\approx \beta p_i^S(t) \sum_j A_{ij} p_j^I(t) \qquad (26)$$

## B.5 PROOFS

**Proposition 2.** *Reaction-diffusion dynamics on graphs is structurally equivalent of the message-passing neural network ansatz.*

*Proof:* Analyzing the full stochastic model requires closely tracking the individual events and varies in each run. Hence, we will work with mean-field diffusion dynamics using transition probabilities, instead. Denoting $p_i^\mu(t) \equiv P(x_i^t = \mu)$ of node $i$ being in states such as $\mu \in \{S, I, ..., R\}$ at time $t$, a Markovian reaction-diffusion dynamics can be written as

$$p_i^\mu(t+1) = \sigma \left( \sum_j F\Big( \mathcal{A}_{ij} \cdot h(p_j)^\mu \Big) \right), \qquad h_a(p_i)^\mu = \sigma \left( \sum_\nu W_{a,\nu}^\mu p_i^\nu + b^\mu \right) \qquad (27)$$

where $\mathcal{A}_{ij}^a = \theta(A_{ij}) f(A)_{ij}$ with $\theta(\cdot)$ being the step function and $\sigma(\cdot)$ a nonlinear function. To see this, note that RD processes on graphs involve a message-passing (MP) step (e.g. an infection signal coming from neighbors of a node), and a reaction step where messages of different states $\mu$ passed to node $i$ interact with each other on node $i$. RD dynamics such as the SIR and SEIR models are also Markovian and the probability $p_i^\mu(t)$ only depends on the probabilities at $t-1$. These are also the conditions satisfied by MPNN. In (27), $\mathcal{A}$ are a set of propagation rules for the messages, which are only nonzero where $A$ is nonzero, same as the aggregation rule in MPNN. To have interactions between states $\mu$ occurring inside each fixed node $i$, $h(p_i)$ can mix the states $\mu$ but not change the node index $i$, leading to the form of $h(p_i)$ in (27), which is the general ansatz for a neural network with weight sharing for nodes, same as in MPNN, and graph neural networks in general. $\square$

### B.5.1 Proof of Theorem 3

**Theorem 3** (Time Horizon). *Assume SIR dynamics* (1) *on a connected graph of $N$ nodes, starting with a single patient zero. Denoting the adjacency matrix by $A$ and its largest eigenvalue by $\lambda_1$, the average infection probability, both over nodes and choice of patient zero, $\langle I(t) \rangle \equiv \langle \sum_i I_i(t)/N \rangle_{\mathrm{P0}}$ becomes $O(1)$ after $t_{\max}$ time steps given by*

$$t_{\max} \sim \frac{\log N}{\gamma(R_0 - 1)}, \qquad R_0 \equiv \frac{\beta \lambda_1}{\gamma} \qquad (28)$$

*Proof:* Consider the spectral expansion $A = \sum_{a=1}^N \lambda_a \psi^{(a)} \psi^{(a)T}$, with $\lambda_1 > \cdots > \lambda_N$. In (1) early in the disease spreading, all nodes are susceptible, meaning $S_i \approx 1$, $R_i \approx 0$, and $I_i \approx 1 - S_i$. Thus, combining the three SIR equations, keeping only $I_i$, the infection spreads as Newman (2018)

$$I_i(t) \approx \sum_j \exp\left[t(\beta A - \gamma \mathbf{I})\right]_{ij} I_j(0) \approx \exp\left[(\beta \lambda_1 - \gamma)t\right] \left( \psi^{(1)} \cdot I(0) \right) \psi_i^{(1)}, \qquad (29)$$

Here, $\mathbf{I}$ is the identity matrix, $\lambda_1$ is the largest eigenvalue of $A$ and $\psi^{(1)}$ is the corresponding eigenvector. Averaging over a uniform choice of patient zeros, for the average infection probability we have

$$\langle I(t) \rangle \approx \frac{1}{N} \exp\left[(\beta \lambda_1 - \gamma)t\right] \left\langle \psi^{(1)} \cdot I(0) \right\rangle_{\mathrm{P0}} \sum_i \psi_i^{(1)} \geq \frac{1}{N} \exp\left[(\beta \lambda_1 - \gamma)t\right] \qquad (30)$$

where we used the inequality between $L_1$ and $L_2$ norms to get $\left\langle \psi^{(1)} \cdot I(0) \right\rangle_{\mathrm{P0}} = \left\| \psi^{(1)} \right\|_1 \geq \left\| \psi^{(1)} \right\|_2 = 1$. Connectedness means $A$ is irreducible and by the Perron-Frobenius theorem its leading eignvector is positive, hence $\sum_i \psi_i^{(1)} = \left\| \psi^{(1)} \right\|_1 \geq \left\| \psi^{(1)} \right\|_2$. Setting the lower bound of (30) equal to 1 and solving for $t$ we get (28). $\square$

### B.5.2 Proof of Theorem 2 $P_{tri}$

*Proof:* If P0 is in a triangle, we may miss it $2/3$ of the times. Thus, the probability of detecting P0 is bounded by $P < 1 - P_{tri} \times 2/3$, where $P_{tri}$ is the probability that P0 is in a triangle. Since edges in $G$ are uncorrelated, each having probability $p$, $G_I$ is also a connected random graph with the same edge probability $p$. Hence, in $G_I$ all nodes have degree $k \approx p|G_I|$. $P_{tri}$ is one minus the probability that none of the $k$ neighbors of P0 are connected, i.e. $P_{tri} = 1 - (1-p)^{\binom{|G_I|p}{2}}$, which proves the proposition. $\square$

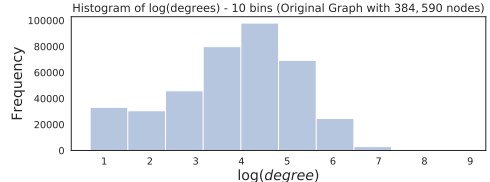
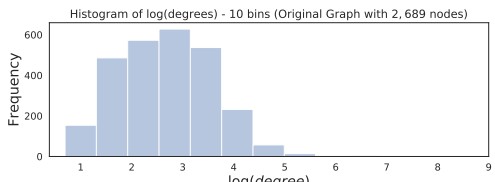

(a) Degree distribution of original network      (b) Degree distribution of subsampled network

Figure 7: Degree distribution of the original co-location network with $384,590$ nodes and the subsampled network with $2,689$ nodes. We subsample the larger network to find a subgraph in order to reduce computational costs of our experiments. We observe that the distribution of our subsampled network is similar to the original graph.

## C   COVID-19 DATA AND SIMULATIONS

**Geolocation data**   Mobility data are provided by Cuebiq, a location intelligence and measurement platform. Through its Data for Good program (`https://www.cuebiq.com/about/data-for-good/`), Cuebiq provides access to aggregated and privacy-enhanced mobility data for academic research and humanitarian initiatives. These first-party data are collected from users who have opted in to provide access to their GPS location data anonymously, through a GDPR-compliant framework. Additionally, Cuebiq provides an estimate of home and work census areas for each user. In order to preserve privacy, noise is added to these "personal areas", by upleveling these areas to the Census block group level. This allows for demographic analysis while obfuscating the true home location of anonymous users and preventing misuse of data.

**Colocation network**   The method for constructing the co-location graphs is as follows. First, we split each day into five minute time windows, resulting in 288 time bins per day. For every location event, we use its timestamp to assign it to a time bin, then assign the longitude-latitude coordinate of the observation to an 8-character string known as a *geohash*. A geohash defines an approximate grid covering the earth, the area of which varies with latitude. The largest dimensions of an 8-character geohash are 38m x 19m, at the equator. If a user does not have an observation for a given time bin, we carry the last observation forward until there is another observation. We finally define two users to be co-located — and therefore to have a timestamped edge in the graph — if they are observed in the same geohash in the same time bin. Accordingly, our co-location graph is constructed by observing the greater Boston area over two weeks from 23 March, 2020 to 5 April, 2020 and results in a graph with $N = 384,590$ nodes. To reduce computational costs, we sample a subgraph with $N = 2,689$ nodes and $|E| = 30,376$ edges with similar degree distribution and connectivity patterns as the original graph and can be observed in Fig 7.

**Epidemic simulations in real data.**   We run a SEIR model on the real co-location network. In doing so, we select parameters and modify the structure of the model to resemble the natural history of COVID-19 Chinazzi et al. (2020). At each time step nodes, according their health status, can be in one of five compartments: $S$, $E$, $I$, $I_a$, or $R$. Thus, we split infectious nodes in two categories. Those that are symptomatic ($I$) and those that are asymptomatic ($I_a$). The first category infects susceptible node, with probability $\lambda$ per contact. The second category instead with probability $r_a\lambda$. We set $r_a = 0.5$ and consider that probability of becoming asymptomatic as $p_a = 0.5$. The generation time, that is the sum of incubation ($\alpha^{-1}$) and infectious period ($\gamma^{-1}$), is set to be $6.5$ days. Specifically, we fix $\alpha^{-1} = 2.5$ and $\gamma^{-1} = 4$ days. In a single, homogeneously mixed, population the basic reproductive number of such epidemic model is $R_0 = (1 - p_a + r_a p_a)\beta/\gamma$ where $\beta$ is the per capita spreading rate Keeling & Rohani (2011). Here however, the epidemic model unfolds on top of the real co-location network. Hence, infected nodes are able to transmit the disease only via contacts (with susceptible individuals) established during the observation period. As mentioned above, the value of $R_0$ is defined by the interplay between the disease's parameters as well as the structural properties of the network Pastor-Satorras et al. (2015); Masuda & Holme (2017). For simplicity we approximate $\beta = \langle k \rangle \lambda$, where $\langle k \rangle$ is the average number of connections in the network. We obtain $\lambda = 0.073$ after solving for $R_0$ and plugging in $\langle k \rangle = 30376/2689 = 11.29$. The simulations start

with an initial infectious seed selected uniformly at random among all nodes. We then read and store the time-aggregated network in memory. The infection dynamics, which are catalysed by the contacts between infectious and nodes, take place on such network. The spontaneous transitions instead (i.e. transition from $S$ to $E$ and the recovery process), take place independently of the connectivity patterns. After the infection and recovery dynamics, we print out the status, with respect to the disease, of each node. Finally, we create a dataset with $10,000$ samples and an $80 - 10 - 10$ train-validation-test split.

