# OpenReview forum: "Finding Patient Zero: Learning Contagion Source with Graph Neural Networks"
_ICLR.cc/2021/Conference — Reject_

### Official Review · AnonReviewer2 · 2020-10-28
**Nice results on the application of Graph Neural Networks to Contagion Source**

**Rating:** 7
**Confidence:** 4

**Review:**

Summary: Backtracking source of an epidemic (Patient Zero (P0)) is one of the important research topics of the current era that helps efficient resource allocation. Many of the existing works in this domain use graph-theoretic measures or message passing algorithms to tackle this problem. In contrast, this paper uses recently emerged Graph Neural Networks (GNNs) to learn and efficiently locate P0. It models disease spreading as a contagion process over a graph. While considering cyclic graphs, the paper shows upper bound on the accuracy of finding P0 and presents a bound of time horizon after which inference become difficult. Experimental results on different real-world and synthetic networks show interesting results.

Comments: The framing of P0 in terms of learning problem and the proposal of GNNs for solving it is a valuable contribution. Also, the theoretical bounds of time horizon and accuracy is quite interesting. Furthermore, Proposition 1 shows an interesting observation that Reaction Diffusion Dynamics is structurally like Message Passing Neural Networks. Such mapping may open new ways to study information diffusion over networks in a learning perspective. Overall, this paper is well-written and structured and friendly to reviewers. I would suggest the following comments for further improving the quality of the paper

1.	Although the experimental details are given in section 4.1 and Appendix A.1. However, it would be useful to add datasets tagging details that how nodes in each simulation were tagged for training GNNs.
2.	Since all the experimental results are based on the randomly chosen P0 and S(E)IR simulations parameters. I encourage the authors adding an ablation study. It would be more helpful to see results on different combination of parameters. Also, the statistical properties such as degree, the number of triangles it lies etc., of P0 would be helpful to see.
3.	To ensure that GNNs work as intended, it would be helpful to see learning curves of GNNs on each type of random graph
4.	 I encourage the authors to release their code and pre-trained models in a format that is easy to reuse for other researchers
5.	The use of \citet and \citep has been mixed throughout the draft. The authors are encouraged to revise the citations accordingly.
6.	The introduction section is lacking in terms of problem motivation. It’s mentioned that the problem is hard however, there is no discussion on the computational complexity of existing graph-theoretic methods and the challenges that need to be addressed.
7.	Eq. 3 \alpha is undefined.

---

### Official Review · AnonReviewer3 · 2020-10-28
**Applying GNNS to finding patient zero**

**Rating:** 3
**Confidence:** 5

**Review:**

An S(E)IR epidemics propagates on a graph, and the goal is to detect its source (P0) only from the observation of the state (S,E,I,R) of every node of the graph at some time $T > 0$. This version of the source detection problem has been studied first by Shah and Zeman (2011) for SI epidemics, as listed in Section 2. The current paper claims to (i) establish new fundamental limits on this problem, showing in particular that after some time the source detection becomes difficult, and (ii) to demonstrate the ability of graph convolutional networks to solve the problem and validate the results on real data.

(i) The theoretical part consists in two short theorems, but their proofs are problematic.

- Theorem 1 builds on an approximate analysis in Newman (2018), which requires a number of assumptions and approximations (for example, a mixing assumption that allows to replace expectations by ensemble averages in ordinary differential equations describing the evolution of $S$, $I$ and $R$; the limitation of time $t \rightarrow 0$). These assumptions and approximation might not be valid on all graphs (e.g., a line graph has the largest eigenvalue around 2, but it is difficult to imagine that $O(1)$ nodes get infected in $\Theta(log(N))$ time). The proof cannot rely on implicit assumptions and approximations whose error is not rigorously estimated. Maybe the theorem is true for some class of graphs including Erdős-Rényi graphs, but even in that case a more involved proof is needed, since the current techniques only work for early times $t$ and not for large times as in (5). Some computations could be clarified, for instance if $ \langle \psi^{(1)} \cdot I(0) \rangle $ denotes the average (over the uniform prior of patient zero over the $N$ nodes) of the scalar product between $\psi^{(1)}$ and the one-hot vector $I(0)$, why is it equal to $\lVert \psi^{(1)} \rVert_1 $ instead of
$ \langle \psi^{(1)} \cdot I(0) \rangle = \frac{1}{N} \sum_{i=1}^{N} \left(  \psi^{(1)} \cdot I(0) \right)_i = \frac{1}{N} \lVert \psi^{(1)} \rVert_1$ ?

- The proof of Theorem 2 is flawed. It starts with the statement that "if P0 is in a triangle, we may miss it $2/3$ of the times." Why would all nodes of a triangle be equally likely to be the source? With the same rationale, why could not we argue that if P0 is in an edge, we may miss it $1/2$ of the times, which would give an upper bound of $1/2$ (as long as $|G_I|>2$) and would contradict the simulations results? The next statement that `"in $G_I$ all nodes have degree $k \approx p|G_I|$" appears incorrect: suppose that the graph is an E-R graph $G(N,p)$, that $R_0$ is very large and that the infection spreads in a snowball way. Then most of the early infected nodes in $G_I$ have node degree $pN$, and not $p|G_I|$.

- Now, the fact that P0's detection becomes harder over time is an important message, especially these days, but it is not a surprising result that when the infected set is a constant fraction of the population, then it is hard to detect P0. This difficulty was already reported in the initial paper by Shah and Zaman (2011).

- In Section 3, the authors contend that compared to the SI model, the removed state introduces additional uncertainty about the temporal order of infections. Why? Since the state of each node is known, having 3 classes (S,I,R) instead of two (S,I) gives more information, which should ease the task of detecting P0.

(ii) The comparison of the GNNs used by the authors over state-of-the-art message passing algorithms is made only with the DMP method of Lokhov at al (2014), but not with the (in general more accurate) belief propagation method of Altarelli et al (2014), also cited in Section 2. There is no comparison with the rumor centrality method developed by Shah and Zaman (2011) either, in terms of accuracy and speed. Also, it would be interesting to see the comparison at times other than $T=30$.

- In terms of speed, training and inference should clearly be separated. It may be misleading to report in the abstract that GNNs are 100 times faster than state of the art methods: that applies only for inference. It would be more accurate to report that GNNs are 100 times faster for inference and twice faster for training compared to the DMP method (as well described in Section 5.1).

- It may be unrealistic to assume that all 4 states (S,E,I,R) can be detected for each individual; the exposed state in particular might be very hard to detect. Otherwise the simulations on real data seem well-done. It is unfortunate that with the non-interpretable theoretical results, the simulation results do not give much insight either. For example, it would be interesting to compare the accuracy results to the size of the infected set on the simulations. Maybe it would be more meaningful to normalize the rank by $|G_I|$ instead of $N$.

- In Figure 2, how are the theoretic curves computed? Equation (7) in Theorem 2 depends on $|G_I|$, which is not directly linked to $T$ nor to the epidemic parameters. Is $|G_I|$ computed based on the simulation results? If so, why are then the confidence intervals not given for these curves?

- In Figures 4 and 5, how is the size of the set $G_I$ evolving over time?

- The paper should be proofread, it contains quite a few typos or vague statements, for instance: Theorem 3 in the appendix is actually Theorem 1 in the main paper; Figure 2 caption does not read well (no verb in the sentence : While accuracy drops below...);  Figure 4 caption: cycles significantly reduces accuracy of P0 -> cycles significantly reduce the accuracy of the detection of P0; Bottom of p4: where each edge has independent an probability $p$ -> where each edge has an independent probability $p$; etc

---

### Official Review · AnonReviewer1 · 2020-10-29
**Timely application**

**Rating:** 5
**Confidence:** 4

**Review:**


This paper proposes to use a graph neural network to infer the source of an epidemic in a network. Given a snapshot of the epidemic, the goal is to determine patient zero without full information of the mechanics of the epidemics but rather learning from historical data.

Overall evaluation:

My evaluation is borderline. Although the application is timely and of interest, there seems to be little methodological novelty (simply an application of an existing GNN architecture) and, for being an empirical paper, the numerical experiments are not entirely convincing (more details below). The theoretical results add to the contributions.


Pros:

1 - Timely application of GNNs to a network problem of interest.

Neutral:

1 - The theoretical statements add to the contributions, although quite simple in essence. For example, Theorem 2 is computing the probability that a given node belongs to a triangle in an ER graph. Proposition  1 seems to be more of an observation than a formal statement. Indeed, Reaction-Diffusion dynamics and MPNN are both non-linear processes on networks. It is unclear if Proposition 1 is saying anything more fundamental than this.


Cons:

1 - No methodological innovation or domain knowledge embedded in the design of the architecture. The authors talk about “several modifications” but these seem to be more accessories (skip connections and batch normalizations) than incorporation of expertise in the architecture or the loss.

2 - For a heavily empirical paper, the experiments are not comprehensive enough. Generalizability across graph types, real-data experiments (instead of simulated data on a real graph), learning from an epidemic mechanism and testing on another one, and learning with just a few observations (how realistic is to observe 20,000 epidemic spreads on the same graph?) are some of the areas that one would expect an empirical paper to cover and are missing in the current version of the manuscript.

---

### Official Review · AnonReviewer5 · 2020-11-10
**Not clear that a ML approach makes sense here**

**Rating:** 3
**Confidence:** 3

**Review:**

This paper studies the problem of source detection in an epidemics when one observes the underlying graph and a snapshot of the population at a given time i.e. who is infected or not infected. For a SIR (or SEIR) model, the authors propose to use GNN for this task. The learning procedure is then the following: given a fixed graph G, the authors create a dataset of snapshots by running a SIR on G.

I am not convinced this problem should be solved with a machine learning approach. In most practical cases, we only have access to one snapshot for a given graph and learning is impossible. The authors here solve this issue by simulating many SIR processes but techniques like the one described by Shah and Zaman without any learning seem much more appropriate.

The authors should compare their results to the results obtained by Shah and Zaman.

[No rebuttal given by the authors] Score unchanged.

---

### Decision · Program_Chairs · 2021-01-07
**Final Decision**

**Decision:**

Reject

**Comment:**

The paper introduces a GNN approach to solve the problem of source detection in an epidemics. While the paper contains some interesting new ideas, the reviewers raised some important concerns about the paper and so the paper should not be accepted in the current form. In particular,

- the paper does not motivate the ML approach to the problem
- the experiments are limited for an empirical paper
- the method used in the paper is not very novel
- the proofs presented in the paper are not formal enough